# Phenolic Extraction of *Moringa oleifera* Leaves in DES: Characterization of the Extracts and Their Application in Methylcellulose Films for Food Packaging

**DOI:** 10.3390/foods11172641

**Published:** 2022-08-31

**Authors:** Fazia Braham, Luísa M. P. F. Amaral, Krzysztof Biernacki, Daniel O. Carvalho, Luis F. Guido, Júlia M. C. S. Magalhães, Farid Zaidi, Hiléia K. S. Souza, Maria P. Gonçalves

**Affiliations:** 1Département des Sciences Alimentaires, Faculté des Sciences de la Nature et de la Vie, Université de Bejaia, Route Targa Ouzemour, Bejaia 06000, Algeria; 2REQUIMTE/LAQV, Departamento de Engenharia Química, Faculdade de Engenharia, Universidade do Porto, Rua Dr. Roberto Frias, 4200-465 Porto, Portugal; 3REQUIMTE/LAQV, Departamento de Química e Bioquímica, Faculdade de Ciências, Universidade do Porto, Rua do Campo Alegre 687, 4169-007 Porto, Portugal; 4IPC—Institute for Polymers and Composites/I3N, Department of Polymer Engineering, University of Minho, Campus de Azurém, 4800-058 Guimarães, Portugal

**Keywords:** methylcellulose films, deep eutectic solvents, *Moringa oleifera* leaf extracts, water vapor permeability, mechanical and optical properties, HPLC-DAD-ESI-MS^n^ analysis, antioxidant capacity

## Abstract

In this work, a qualitative study of the phenolic content of *Moringa oleifera* leaves (MO), extracted with deep eutectic solvents (DES) based on choline chloride (ChCl) with lactic acid (LA) or glycerol (GLY), was performed by high-resolution mass spectrometry (HPLC-DAD-ESI-MS^n^). The two solvents (DES-LA and DES-GLY) extract similar classes of phenolics, and ten compounds were identified. The antioxidant profile was also studied (TPC, TFC, DPPH, FRAP, ORAC, and ABTS). Both solvents show an efficient extraction of phenolic compounds and high antioxidant capacity was verified for the extracts. However, the DES-Gly have a higher capacity for polyphenolic extraction (TPC led to 38.409 ± 0.095 mg GAE.g^−1^ and 2.259 ± 0.023 mg QE.g^−1^ for TFC). Films based on methylcellulose (MC) containing different amounts of DES or MO extracts, acting as plasticizers, were developed and characterized regarding their mechanical, optical, water vapor permeability, and microstructural properties. All films are uniform, clear, and transparent with smooth, homogeneous surfaces. It was found that the presence of more than 10% of MO extract and/or DES provided more flexible films (Eb for MC 2%_DES 20% achieved 4.330 ± 0.27 %, and 8.15 ± 0.39 % for MC 2%_MO 20%) with less mechanical and barrier resistance. The ultimate objective of this study was to provide information that could assist in the development of antimicrobial active methylcellulose films for sliced wheat bread packaging.

## 1. Introduction

The growing demand of consumers for minimally processed foods, associated with a swing away from the addition of synthetic preservatives used to prolong the microbial and sensory life of bakery products, and the recent request for environmentally friendly packaging, encouraged the search for innovative solutions to extend the shelf lives of these products. Natural extracts (i.e., polyphenol and oils) have been used as antimicrobial additives for these purpose. In particular, the use of *Moringa oleifera* to improve the efficiency of food production systems, nutrition, and human health has attracted the interest of numerous researchers. Since its components, mainly in the leaves, consist of carotenoids, tannins, vitamins, flavonoids, and phenolic acids presenting potent antimicrobial and antifungal activities, it is considered one of the most useful plants in the world to be used in different applications [1,2,3].

Traditionally, conventional organic solvents with liquid–liquid extraction or solid-phase extraction were applied to extract phenolic compounds from plants. However, from the viewpoint of green chemistry, most of the commonly used solvents are volatile, flammable, and toxic liquids. Besides, these extraction methods have the major disadvantage of solvent residue being present in the extracts. In this sense, an innovative category of solvents such as Deep Eutectic Solvents (DES) has been used to overcome the disadvantages of organic solvents. In recent years, sustainable “green” extraction procedures have been published for the recovery of phenolic compounds from *Moringa oleifera* [4,5,6].

Edible and coating films derived from natural polymers have been used to protect the quality and extend the shelf life of food products. In this sense, there are few studies in the literature regarding the use of *Moringa oleifera* in polymer-based food packaging. Lee et al. [7] recently developed an antimicrobial, antioxidative, and biodegradable packaging material from puffer fish skin gelatin containing *Moringa oleifera* leaf extract. Núñez-Gastélumn et al. [8] determined the physical–structural properties of biodegradable films of chitosan and polycaprolactone (PCL) added with *Moringa oleifera* leaf extract. Tesfay & Magwaza [9] reported the potential of edible coating containing *Moringa oleifera* leaf extracts to improve avocado fruit quality during the storage period. More recently, Bhat et al. studied [10] the incorporation of *Moringa oleifera* leaf extract into a chitosan/guar gum/poly(vinyl alcohol) matrix to produce active films. Also, triggerfish gelatin films containing *Moringa oleifera* leaf were used to package ricotta cheese [11].

An emerging area of food technology is active packaging, where absorbing and/or releasing compounds that are effective against the aging of bread and/or with antimicrobials, are incorporated into the packaging itself, thus preventing the growth of undesirable microorganisms. The major changes that occur after baking are moisture redistribution, starch retrogradation, increased firmness, and loss of aroma and flavor. Staling, in which the starch begins to expel excess water returning to its crystalline form, is a spontaneous and irreversible phenomenon, and firming can easily be controlled by active packaging [12]. Thus, the development of surface molds is the main reason for the shelf life reduction of bakery products. The growth of microorganisms in bread-making products begins or is more intense at the surface, which makes the use of edible films and coatings a good choice for maintaining the quality of these products [13]. Agents may be incorporated into the packaging that can directly interact with spoilage organisms or interact with the environment within the packaging without the direct addition of chemical agents [14,15].

Cellulose is one of the most abundant agricultural biopolymer wastes produced on the earth. Thus, this polysaccharide is useful for the creation of new approaches for various commercial food applications [16,17,18]. For example, cellulose-based edible films have good barrier properties (aroma, oxygen) and oil transfer similar to other hydrophilic films. Moreover, strong and flexible water-soluble films were produced by cellulose derivatives, such as methylcellulose and hydroxypropyl methylcellulose, and may offer a promising alternative to transport natural antimicrobial agents [19]. In this regard, previous studies using cellulose and its derivatives have shown that the microbial shelf life of bread can be increased by active packaging. Vojdani & Torres [20,21] have examined the diffusion barrier properties of methylcellulose and hydroxypropyl methylcellulose films with great potential to be used as a substrate for the antimicrobial agent potassium sorbate. The addition of fatty acids (e.g., lauric, palmitic, stearic, and arachidic acids) have been demonstrated to diminish the diffusion of potassium sorbate in cellulose-based films [22]. More recently, Otoni et al. [23] successfully produced methylcellulose films with the incorporation of micro and nanoemulsions of clove bud and oregano essential oils that displayed antimicrobial activity against spoilage fungi in bakery products. Noshirvania et al. [24] studied a nanocomposite film and coating, based on chitosan-carboxymethyl cellulose-oleic acid incorporated with different concentrations of zinc oxide nanoparticles, as a high barrier packaging material acting as a preservative (microbial and staling) on the surface of sliced wheat bread.

This work aims to evaluate the in vitro effectiveness of bio-based films prepared from methylcellulose with and without incorporating deep eutectic solvent (DES) and *Moringa oleifera* leaf extracts (MO) and to ascertain its applicability in the design of active food packaging systems for wheat sliced bread.

## 2. Materials and Methods

### 2.1. Materials

Calcium chloride ≥ 97.0%, (CAS: 10043-52-4), granular drying agent, and choline chloride ≥ 99.0%, (CAS: 67-48-1), were obtained from Sigma-Aldrich (Darmstadt, Germany), glycerol ≥ 99.5%, (CAS: 56-81-5) was purchased from Panreac, DL-lactic acid (≥90%) from Sigma-Aldrich, *Moringa oleifera* leaves were harvested and treated as described previously [25], and Methocel^TM^ (Methylcellulose, CAS: 9004-67-5, D.S. 25–33%), was purchased from SpecialIngredients^®^. Methanol (purity 98%) was purchased from Panreac, and all the other reagents used for antioxidant capacity were of analytical grade and purchased from Sigma-Aldrich.

### 2.2. Methods

#### 2.2.1. DES Preparation

The DES used in this study were prepared as previously described [26]. Briefly, choline chloride (ChCl), previously dried at 70 °C overnight in an oven and used as quaternary ammonium salt (HBA), was mixed with an appropriate hydrogen bond donor (Lactic acid (LA) and Glycerol (GLY)) at molar ratio 1:2.

A homogeneous liquid was formed by heating the mixture at 80 °C under constant stirring. After that the solution was left to cool at room temperature. The water content for each DES (in triplicate) was determined by Karl Fischer titration using a Metrohm Karl Fischer coulometer and it was found to be lower than 2% for all the DES.

#### 2.2.2. Extraction of *Moringa oleifera* Leaves in DES

Phenolic compounds of *Moringa oleifera* leaves were extracted in deep eutectic solvents DES (ChCl/GLY 1:2) or (ChCl/LA 1:2) following a previously published methodology by Oomah et al. [27]. In brief, 1 g of *Moringa oleifera* leaves flour was mixed with 40 mL of DES and extracted by continuous stirring at 900 rpm for 2 h. Then, the extract was centrifuged for 30 min, at 11,000 g, and the supernatant was carefully removed and stored in the freezer (−18 °C).

#### 2.2.3. Spectrophotometric Analysis of *Moringa oleifera* Leaf Extracts

##### Phenolic Compounds

The total phenolic content (TPC) was estimated by the Folin–Ciocalteu standard method as previously reported by Skerget et al. [28]. Briefly, 2.5 mL of Folin–Ciocalteu reagent (10% *v*/*v*) was added to 0.5 mL of diluted extract followed by 2 mL of sodium carbonate (75 g.L^−1^) after 5 min. The samples were incubated at 50 °C for 5 min then cooled and the absorbance was measured at 760 nm. TPC was expressed as mg gallic acid equivalent (GAE) per g of sample.

Total flavonoid content (TFC) was measured according to Djeridane et al. [29]. 1 mL of 2% AlCl_3_.6H_2_O (2 g in 100 mL methanol) solution was added to the same volume of diluted extract, then vortexed and left for 15 min in the dark at room temperature. Afterwards, the absorbance was monitored at 430 nm and TFC was expressed as mg quercetin equivalent (QE) per g of sample. All the analyses were performed in triplicate.

##### Antioxidant Capacity

The determination of the antioxidant capacity of *Moringa oleifera* leaf extracts was carried out following DPPH, ABTS, FRAP, and ORAC assays. All measurements were performed in triplicate and expressed as mmol Trolox equivalents (TE) per g of sample, except ABTS which was expressed as percentage of inhibition (PI).

A microplate reader (BioTek Instruments Inc., San Diego, CA USA) was used to perform DPPH and FRAP assays according to the previously established protocol [30]. In brief, 20 µL diluted extracts, blank (solvent), and standard were introduced into microplates and 300 µL DPPH solution freshly prepared (10 mg in 250 mL of methanol) was added. The samples were placed in the dark for 30 min at room temperature then the absorbance was monitored at 515 nm. The ability of *Moringa oleifera* leaf extracts to reduce the complex of Fe^3+^-2,4,6-tris(2-pirydyl)-s-triazine into Fe^2+^-2,4,6-tris(2-pirydyl)-s-triazine was determined by adding 300 µL fresh FRAP reagent to 50 µL of diluted extracts and the absorbance was read at 593 nm after 5 min.

The oxygen radical absorption capacity assay (ORAC) was determined according to Huang et al. [31], with some modification as already detailed in our previous work [25]. The ABTS radical scavenging capacity was measured following the method of Giao et al. [32], in which a solution containing 7 mmol.L^−1^ ABTS (2,2-azinobis(3-ethylbenzothiazoline-6-sulfonic acid) diammonium salt) and 2.45 mmol.L^−1^ potassium persulfate in equal volume was allowed to react in the dark for 16 h at room temperature. This solution was diluted until an absorbance of 0.700 ± 0.02 was reached at 735 nm. 20 µL of the extract was mixed with 1.5 mL of the prepared solution and the absorbance was measured after 6 min at 735 nm. The absorbance of the solvent was subtracted from that of the sample in all tests.

#### 2.2.4. HPLC-DAD-ESI-MS^n^ Analysis

The phenolic profile of Moringa oleifera leaves extracted with deep eutectic solvent (GLY and LA) was studied using the HPLC system (Thermo Electron Corporation, Waltham, MA, USA) coupled with an ion-trap mass spectrometer and diode array detector (DAD), equipped with a low-pressure quaternary pump with an autosampler and a diode array detector (Finnigan Surveyor Plus, Thermo Fisher Scientific, Waltham, MA USA). The qualitative analysis of the samples was carried out as described previously.

#### 2.2.5. Films Preparation

For the determination of the proper concentration of methylcellulose, (MC), the flow properties of different concentrations of MC were studied using 1–2 and 5% (dry basis) in distilled water. A rheometer (TA Instrument, New Castle, DE, USA) was used to measure the viscosity of these solutions. The solution that presented the best properties was MC 2%, since for higher concentrations of MC the solutions have a very high viscosity.

The MC solution was prepared by first dispersing the MC powder with 1/3 of the total required volume of distilled water that had been heated to above 70 °C. For complete solubilization, the remainder of the water was then added as cold water to lower the temperature of the dispersion. Stirring continued until all particles had been thoroughly wetted. The solution was then stored overnight at T = 4 °C to remove all bubbles.

Three samples of about 1 g each of MC solutions were dried at T = 105 °C and weighed to ±0.0001 g for the determination of the MC dry mass.

Film-forming solutions preparation: DES (ChCl:glycerol, 1:2), or DES *Moringa oleifera* extract (MO) were added as the plasticizer to the MC solution, and the composition of various films used is given in Table 1.

The films were produced in different plasticizer proportions in order to find the formulation that resulted in films with better performance.

Films were made by the knife-coating technique (Sheen, model 1132N, Cambridge, UK). Seventy grams of the solution was spread at 0.3 m·s^−1^ on acrylic plates using an 1132N film applicator (Sheen Instruments, Cambridge, UK). The films were dried in a climate chamber (Binder KMF 115) for 24 h (40 °C and 60% relative humidity (R.H.)), removed from the acrylic plates, and then conditioned at 25 °C and 53% R.H. for at least 72 h before performing their characterization.

### 2.3. Film Properties

#### 2.3.1. Thickness Measurements

The thickness of the films was measured using a digital micrometer (Mitutoyo Co., Japan, model ID-F150). The average thickness was calculated from measurements taken at ten different locations on each film sample.

#### 2.3.2. Water Vapor Permeability (WVP)

The water vapor permeability (WVP) tests were determined gravimetrically as described elsewhere, and according to ASTM Standard Test Methods for Water Vapor Transmission of Materials (E 96-00) [33]. Briefly, the test films (80 mm in diameter) were sealed on top of the permeation cells (Elcometer 5100/2 Payne Permeability Cup), containing anhydrous calcium chloride (R.H. = 2%) and put in a desiccator at a constant temperature of 21 ± 1 °C and R.H. 100%. Air convection of approx. 0.3 m·s^−1^ was used to promote water diffusion (using a fan) and to ensure uniform conditions throughout the chamber. The cells were weighed periodically. The measured WVP of the films was determined as follows:(1)$$\mathrm{WVP}\;=\frac{\Delta m\cdot x}{A\cdot \Delta t\cdot \Delta p},$$
where WVP is the water vapor permeability (g·m^−1^·s^−1^·Pa^−1^) ^1^), Δ*m* is the weight gain (g), *x* is the film thickness (m), and A is the area (0.003 m^2^) exposed for a time Δ*t* (s) to a partial water vapor pressure Δ*p* (Pa).

For each type of film, WVP measurements were replicated three times for each batch of films.

#### 2.3.3. Mechanical Properties

Mechanical properties of the films (tensile strength, TS, elastic modulus, YM, and elongation at break, Eb), were measured using a texture analyzer (TA.XT2, Stable Micro Systems Ltd., Surrey, UK) equipped with adequate tensile test grips (A/TG model). Once conditioned, films specimens with dimensions of 25 mm × 100 mm were mechanically fractured; the applied test speed was 0.2 mm s^−1^, while force (N) and deformation (% strain) were recorded. At least seven replicates were used for each film formulation.

#### 2.3.4. Scanning Electron Microscopy (SEM)

The scanning electron microscope FEI Quanta 400 FEG, under high vacuum (5 kV voltage and ≈11.1 mm working distances—CEMUP—Centro de Materiais da Universidade do Porto-PT) was used to study the morphology of the films. Before analyses, film samples were rapidly cooled in liquid nitrogen, fractured, mounted on aluminum stubs, covered with double-coated carbon conductive adhesive tabs, (Electron Microscopy Sciences, Hatfield, PA, USA), and coated with a thin film of Au-Pd.

### 2.4. Bread Experiences

For many types of bread, the relatively high moisture content of the crumb is responsible for the appearance of mold, with the bread’s shelf life being around 4 to 8 days. Sensory shelf life can be even shorter with some bread brands staling within 24 h although 6–10 days is more common [34].

#### 2.4.1. Evaluation of Antifungal Effectiveness of Methylcellulose Films in Sliced Bread

The wheat bread was purchased from a local bakery, cut into 20 mm thick slices, and wrapped with methylcellulose film formulations: MC 2%; MC 2% _DES10%, and MC 2% _MO10% (active packaging tests), sealed in plastic bags to avoid moisture loss, and left in room temperature, 20 ± 2 °C, to simulate usual commercialization conditions of bakery products. Control packaging tests were performed in the same way without including the MC film. Only if there is a delay in fungal growth, when compared to the control, is the packaging considered effective. Tests were performed in duplicate.

#### 2.4.2. Determination of Bread Firmness

The determination of the bread firmness was carried out using a TA-XT2 (Stable Micro Systems) texture analyzer, equipped with a cylindrical probe of 6 mm (P/6). From the force versus deformation curves, the maximum puncturing force, in N, was obtained. At least 8 measurements were made for each sliced bread.

#### 2.4.3. Weight Loss Measurements

The weight loss of bread was determined gravimetrically and expressed as a percentage of initial bread weight.

### 2.5. Color Measurements

Color was measured with a Minolta colourimeter CR 300 (Tokyo, Japan), using a white standard as a background to determine the CIELAB color parameters, lightness, L* (0 [black] to 100 [white]), and chromaticity parameters a* (green [−60] to red [+60]) and b* (blue [−60] to yellow [+60]). The opacity of the films was also measured using black and white standards. For each sample, the test was performed in several zones, with a minimum of 5 scans for every treatment, and an average value was obtained.

### 2.6. Statistical Analyses

Software package GraphPad Prism San Diego, CA, USA, version 5.00 for Windows, was used for the experimental results evaluation. The mean differences of the film properties were examined by an ANOVA with Bonferroni’s post hoc-test and were considered significant at *p* < 0.05.

## 3. Results and Discussion

### 3.1. Spectrophotometric Analysis of Moringa oleifera Leaf Extract

Deep eutectic solvents offer advantages in the extraction process including low price, chemical inertness with water, ease of preparation, biodegradability, and so on [35]. In addition, their compatibility with food industries allows them to be more appropriate than organic solvents. In this context, two types of DES were tested in the extraction of antioxidants from *Moringa oleifera* leaves. As illustrated in Table 2, both of ChCl/GLY and ChCl/LA extracts show an efficient extraction of polyphenols and exhibit a high antioxidant capacity where values of TPC and TFC attained 38.409 ± 0.095, 28.314 ± 0.146 mg GAE.g^−1^, and 2.259 ± 0.023, 1.595 ± 0.023 mg QE.g^−1^, respectively, also the antioxidant capacity led to 0.361 ± 0.017, 0.469 ± 0.003 mmol TE.g^−1^ for FRAP and 1.200 ± 0.066, 2.107 ± 0.003 mmol TE.g^−1^ for ORAC assay. Statistical data show significant differences between the two solvents, therefore, ChCl/GLY extract presents a higher amount of TPC and TFC, and similarly the highest antiradical potential against DPPH and ABTS (0.294 ± 0.010 mmol TE.g^−1^ and 92.614 ± 0.085%). On the other hand, ChCl/LA extract shows more ability to reduce iron and scavenge peroxyl radicals, and the obtained results are in accordance with a previous study [36], denoting that the ChCl/LA extract exhibits the highest FRAP value and the lowest ABTS value compared to the other DES extract. In addition, the antioxidant power of an extract can be influenced by several factors, including the polarity and solubility of the extracted ingredients, structure (number and the position of hydroxyl groups), the free radicals’ nature and their interaction mechanisms with other moieties, and the solvent type [37].

### 3.2. HPLC-DAD-MS^n^ Profiling of Phenolics from Moringa oleifera Leaf Extracts

The qualitative analysis of the phenolic composition from extracts of *Moringa oleifera* leaves obtained using DES with lactic acid (LA) and glycerol (GLY) has been carried out using HPLC-DAD-ESI-MSn in negative ionization mode. The chromatograms of both extracts recorded at 280 nm are shown in Figure 1. Similar phenolic compounds were extracted with DES-LA and DES-GLY, however, the variation found in terms of antioxidant capacity can be explained by the difference in the relative concentration of extracted compounds. They were tentatively identified by MS data together with the interpretation of MSn and UV spectra in comparison with those found in the literature. The identities, retention times, and observed molecular and fragment ions of individual compounds are presented in Table 3. Ten phenolic compounds were tentatively identified (identified as peaks 1–10).

Two structural isomers of caffeoylquinic acid (CQA) were identified in *Moringa oleifera* leaf extracts (peaks 1 and 3 with retention times of 25 and 40 min). These peaks exhibited a characteristic intense molecular ion with *m*/*z* 353 in negative ion mode and a characteristic absorption maximum at 325 nm (Table 2), characteristic from CQA. CQA yielded a fragment ion at *m*/*z* 191, which produced an MS/MS spectrum identical to that of quinic acid (*m*/*z* 85, 127, 173). Therefore, *m*/*z* 191 is due to quinic acid ion resulting from cleavage of the C-O bond of the ester linkage. The ion at *m*/*z* 179 was identified at the caffeic acid portion of chlorogenic acid [38,39]. The ion at *m*/*z* 179 loses a carbon dioxide moiety to produce the product ion observed at *m*/*z* 135. Peaks 1 and 3 fragment into *m*/*z* 135, 179, and 191 ions. Compound **1**, one of the most intense peaks on the chromatogram of both extracts, was tentatively attributed to 3-CQA, by giving the characteristic negative ion fragments of *m*/*z* 191 and 179, which has also been previously identified as one of the main phenolic compounds in hydroalcoholic and hydroacetonic extracts of *Moringa oleifera* leaves [25]. Peak 3 was identified as 5-CQA by comparison with the reference pure standard, with an MS/MS base peak *m*/*z* 191 and a low-intensity MS/MS ion at *m*/*z* 179 [39]. The 4-CQA can be differentiated from the other isomers by having a base peak at *m*/*z* 173 [39,40]. No similar fragmentation pattern was found in the leaves extract, so the presence of 4-CQA was excluded in these extracts.

Peak 2 (Rt 33 min) showed a pseudomolecular ion at *m*/*z* 337 and was tentatively identified as p-coumarylquinic acid. As previously described, the precursor ion was fragmented in product ions corresponding to quinic acid (*m*/*z* 191, [quinic acid–H]-; and *m*/*z* 173 [quinic acid–H–H_2_O]-), p-coumaric acid (*m*/*z* 163, [p-coumaric acid–H]-), and p-coumaric acid residue (*m*/*z* 119, [p-coumaric acid–H–CO_2_]-) [41].

Compound **4** was identified as the di-C-glucosyl flavone Apigenin 6,8-di-C-glucoside (vicenin-2). The compound showed a deprotonated molecule at *m*/*z* 593, and a characteristic fragmentation pattern with fragment ions at *m*/*z* 575 [(M-H)-18], *m*/*z* 503 [(M-H)-90], *m*/*z* 473 [(M-H)-120], *m*/*z* 383 [(M-H)-90-120], and *m*/*z* 353 [(M-H)-120-120]. The vicenin-2 was recently reported in *Moringa oleifera* leaves [25]. Peak 5 and 6 exhibited a pseudomolecular ion at *m*/*z* 431. The MS/MS spectra gave fragment ions at *m*/*z* 341, 311, and 283, consistent with the presence of vitexin or isovitexin, as previously reported [25]. Peak 7 was identified as Rutin by comparison with the reference standard, and is one of the main phenolic compounds present in the extracts. The mass of the parent ion is 609 and its fragmentation pattern revealed a base peak ion at *m*/*z* 301, corresponding to the deprotonated aglycone. Peaks 8, 9, and 10, very abundant in Moringa oleifera DES extracts, were tentatively identified as kaempferol glycoside derivatives. Peak 8, with a retention time of 72 min and a characteristic absorption maximum at 350 nm, produced a deprotonated molecular ion peak at *m*/*z* 447 and yielded fragment ions at *m*/*z* 327 [(M-H)-120] and *m*/*z* 285 [(M-H)-162], consistent with the loss of a glucosyl reside and the generation of the characteristic aglycone fragment, respectively. Other characteristic fragment ions at *m*/*z* 255 and 227 were also observed (Table 1). The compound was therefore identified as kaempferol-3-O-glucoside, as previously described [42,43], and also identified in *Moringa oleifera* leaf extracts [44]. Peak 9, the most intense peak identified in the extracts at 280 nm, exhibits a deprotonated molecule at *m*/*z* 593. According to the information previously reported by Ibrahim et al. [42] this peak may be attributed to kaempferol-3-O-glucoside-7-rahmonisde. The fragmentation patterns with a characteristic fragment at *m*/*z* 285 is consistent with the loss of glycosidic residues (rhamnosyl, 146 amu; and glucosyl, 162 amu) [42]. Peak 10 was tentatively identified as kaempferol-3-O-malonylglucoside according to the mass spectrum with a deprotonated molecule at *m*/*z* 533 and characteristic absorption maximum at 264 and 346 nm. The molecular ion fragmentation yielded ions with *m*/*z* 489 and 447 related to kaempferol glucoside after the loss of the CO_2_ ([M-H]-44) and subsequent malonyl moiety ([M-H]-86). The presence of the ion at *m*/*z* 285 confirms the presence of aglycone kaempferol after losing a hexose moiety ([M-H]-162).

### 3.3. Color Parameters and Optical Properties of Films

The effect of DES-GLY or MO extract (extracted in DES-Gly) incorporation on the color parameters of the MC films is shown in Table 4. All the films are uniform, clear, and transparent with high lightness (high Hunter L-value of 93). A decrease in lightness (*p* < 0.05) was observed in films with 20% DES or 20% MO extract. The chromatic coordinate parameters increased (red-green (a*)) or decreased (blue-yellow (b*)), significantly by increasing the plasticizer content (20% for DES or any MO content) in relation to the control film (MC 2%). In addition, MC 2% films loaded with 20% DES or 20% MO extract showed higher ΔE* than MC control. Moreover, the opacity values of films plasticized with 10% DES or 10% MO extract were similar to the control film and higher for 20% MO extract. The increase in the opacity for the MC 2%_MO 20% film can be mainly attributed to the presence of Moringa compounds in the formulation.

### 3.4. Thickness, Mechanical Properties, and Water Vapor Permeability of MC Films

Thickness, d, around 0.019 mm was obtained for 2% MC films, while no significant differences (*p* > 0.05) were observed for the addition of MO extract or DESs plasticizers. Similar behavior was observed for the tensile strength, TS, and no significant differences (*p* > 0.05) were found in the rigidity when 10% plasticizer was added to the film formulation. However, the intermolecular bonds between the MC chains are reduced (TS reduction (*p* < 0.05)) when 20% DES or any MO content is added to the formulation, leading to a weaker (less resistant) film. The elasticity, Eb, of the 2% MC films with the addition of DES (10% or 20%) or MO extract (10%) as plasticizing agents showed no significant difference (*p* > 0.05) in relation to the 2% MC films with no plasticizer added. The addition of more MO extract (20%) led to a remarkable plasticizing effect providing an almost 3.5-fold increase in Eb (*p* < 0.05) when compared to the control film. Regarding the YM values, it was observed that the addition of any plasticizer seems to destabilize the stable structure of the MC network, leading to a decrease (*p* < 0.001) in the YM values. In other words, the incorporation of DES or MO extract led to more flexible films, the effect being more pronounced in films plasticized with 20% MO extract, as can be confirmed by the Eb values.

The tensile strength (TS) and Eb values reported in the literature for pure MC films produced by the casting method were higher than our results [45,46,47,48,49]. The differences are probably due to physicochemical characteristics of the methylcellulose used, film production conditions, and different thicknesses obtained. In the last two decades, deep eutectic solvents have been tested as plasticizing agents for polysaccharide films [50]. Choline-based deep eutectic solvents have been found to efficiently plasticize pectin [26], starch [51], guar gum [52], or chitosan films [53,54]. However, to the authors’ knowledge, to date, no data on the mechanical properties of DES-plasticized MC films have been reported, therefore, no comparison with the literature can be made. According to the literature, the addition of different plasticizers (malic acid, sorbitol, PEG400, or glycerol) decreases TS and increases Eb values in MC films [48,49], which is in agreement with the results presented in the Table 5.

Regarding the barrier properties, the water vapor permeability (WVP) values were statistically dependent on the DES or MO extract content (*p* < 0.05) An increase in WVP (hence a decrease in barrier properties) was observed for all MC films containing plasticizer, with the greatest increase being observed for MC films with 10% or 20% MO extract (Table 5). However, the increase of DES or MO extract contents on the MC-film formulations do not affect the WVP values. The decrease in the water vapor barrier properties of the plasticized samples is mainly due to the hydrophilic nature of the DESs and MO extract components.

Similar behavior was observed with Khorasan wheat starch films containing MO extract, the phenolic compounds in the extract affected the interactions between starch and water molecules and caused the increase in WVP [55]. In other studies, the addition of PEGs with several MWs to the MC matrix resulted in an increase in the WVP of the films [48] while the incorporation of resveratrol, a lipophilic compound, tended to reduce the WVP of MC films [49].

### 3.5. Morphologic Analysis of Films

Figure 2 shows SEM micrographs of the surface of the films. The 2% MC film, without plasticizer, was characterized by having a smooth, homogeneous surface (Figure 2a) and compact cross-section (Figure 3a). In corroboration with the TS results, 10% of DES does not affect the surface structure or cross-section of the film. However, the MC films containing 20% DES and MO extract (10% and 20%) showed a relatively loose cross-sectional structure, and some surface irregularities (increased porosity) are observed in the films (Figure 2c–e and Figure 3c–e), which could illustrate the decrease in the strength of the films measured in the mechanical tests carried out. The microstructure of the films observed by SEM was consistent with the results of TS, YM, and WVP (Table 4). In this sense, the 2% pure MC film presented the highest values of TS and YM and the lowest permeability, which corroborates its relatively dense and homogeneous image observed by SEM.

Cross sections of the cryo-fractured MC films observed by SEM are represented in Figure 3.

### 3.6. Effectiveness of MC Films on Sliced Wheat Bread

The shelf life of sliced wheat bread has been investigated. As previously observed, the MC_10% DES film presented properties close to the control film. For this reason, it was selected for further tests performed on a food model. The MC 2%_ MO 10% film was used for comparison. Accordingly, several samples of sliced wheat bread wrapped with MC films were compared with the control (unwrapped sliced wheat bread). The samples were evaluated for the following characteristics: texture, color, visual appearance, and mold growth, during a period of 11-days.

Figure 4 shows the growth of mold on slices of bread exposed to a humid environment, after 6, 7 and 11 days of storage, at room temperature, 20 ± 2 °C. It is well known that the growth of fungi in foods can be affected by a number of chemical and physical parameters (e.g., temperature, water activity, nutrient status, etc.).

As the bread formulation used appeared to contain no preservatives, fungal growth was easily detected in the control bread (Figure 4). Regarding the formulations, MC films seem to delay fungal growth, regardless of the type of film used. A possible explanation could be the fact that the film absorbs some of the water on the surface of the bread making the medium less favorable to mold growth, which is in full agreement with WVP results (Table 5).

These findings are in agreement with previous studies [56,57], which reported that water barrier properties and water vapor sorption by the package had an effect on moisture accumulation and microbial growth, playing an important role in extending bread shelf life.

Firmness (N) values for sliced bread during storage are shown in Table 6. Firmness is one of the key factors in determining bread quality and, in addition to starch retrogradation, distribution of water plays an important role by increasing the moisture content, causing less firmness of the bread [58].

The increase in bread crumb firmness is visible for all samples over time, reaching the highest value for bread with MC_MO. The control sample showed no significant difference in the parameter after 1 day of exposure to the environment. Also, for the first day, the results indicate a higher firmness for bread with MC_MO when compared to the other samples. The migration of moisture from the crumb to the MC_MO film could be the phenomenon responsible for the increase in bread firmness. In addition, after 4 days, the MC 2% film seems to protect the samples, since the bread firmness parameter is not significantly affected compared to the others (MC_DES and MC_MO).

The results of the weight loss of bread slices, measured on days 1, 4, 7, and 11, are shown in Table 7. It appears that weight loss is more pronounced for the control, while a slightly better overall result was obtained for the bread wrapped in the MC and MC_DES films. On the other hand, the bread wrapped in the MC_MO film showed the greatest weight loss of all samples on day 11.

Bread crumb color changes on days 0, 1, 4, 7 and 11 of storage are collected in Appendix A. The color of the breads (control and packaged) was affected by storage, which is consistent with the results reported in the literature [59].

It was observed that the bread crumb color showed a significant increase (*p* < 0.05) in redness (a* value) and yellowness (b* value) but a decrease in L* value after 11 days of storage. The decrease (*p* < 0.05) in the value of L indicates the intensification in the development of the darker color of the bread crumb, compatible with an outcome of Maillard reaction and also associated with the fungal growth in the bread. The increase in a* values could be explained by the loss of moisture during storage.

Minolta color a*/b* values decreased with increasing time storage. The values varied less until the 4th day of storage, with results between −0.1 and −0.12 being obtained for the control sample and the one wrapped in 2% MC film; lower values were observed for samples wrapped in 2% MC films with plasticizer values up to the 7th day also negative, but positive values were found after 11 days of storage. Interestingly, on day 11, the Minolta a*/b* color value is similar for all samples except the one wrapped in MC 2%_DES 10% film.

## 4. Conclusions

In this work, methylcellulose-based active films were prepared with DES and MO extracts at different concentrations to study the effect on the physical properties of the polymer and the possible application as active packaging for sliced wheat bread. At the highest MO concentration, films showed some changes in their physicochemical and microstructural properties. The presence of phenolic compounds affected the interactions between polysaccharides and water molecules and caused the increase in WVP of the films, but did not affect the tensile strength, TS. From the results, it is clear that the incorporation of DES and MO extract influenced attributes such as firmness, water permeability and visual microbial growth in bread. Overall, the films have demonstrated some effectiveness against mold growth, so these active cellulose-based films could be an approach to increasing the microbial shelf life of bread without directly adding chemical preservatives to the baking formula. No conclusions have been drawn as to whether films can slow staling in bread.

## Figures and Tables

**Figure 1 foods-11-02641-f001:**
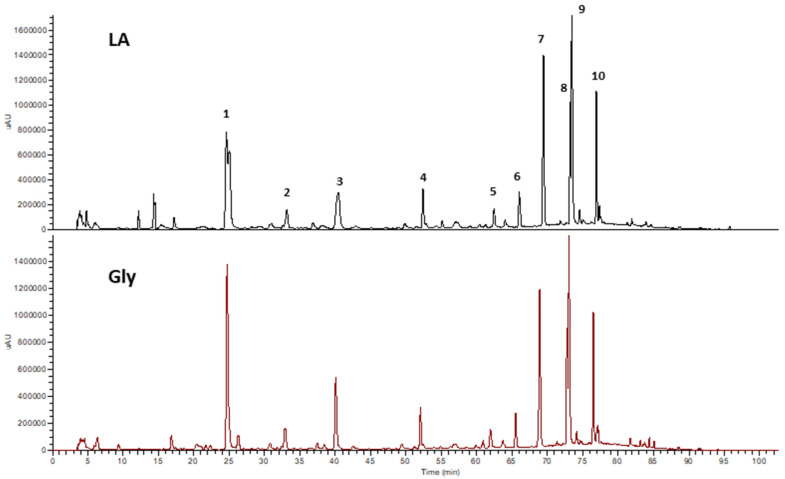
HPLC-UV chromatograms of *M. oleifera* extracts prepared with DES (with LA and GLY) detected at 280 nm. Peaks are tentatively identified in Table 2.

**Figure 2 foods-11-02641-f002:**
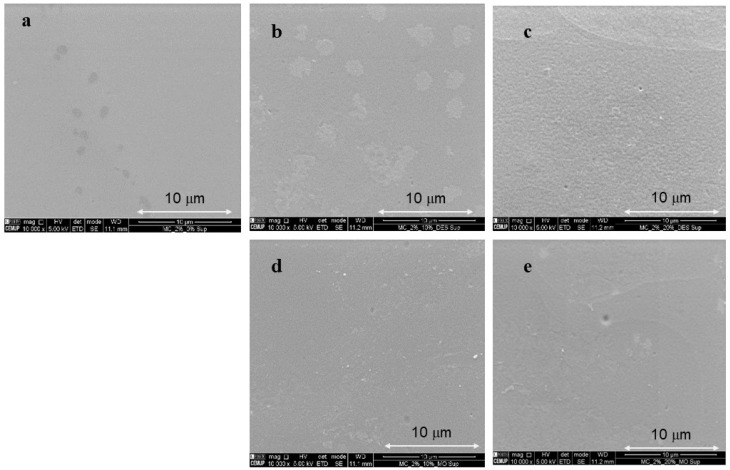
SEM micrographs of all studied film samples (magnification 10,000×, 5.00 kV). (**a**) Pure MC 2%, (**b**) MC 2%_DES 10%, (**c**) MC 2%_DES 20%, (**d**) MC 2%_MO 10%, (**e**) MC2%_MO 20%.

**Figure 3 foods-11-02641-f003:**
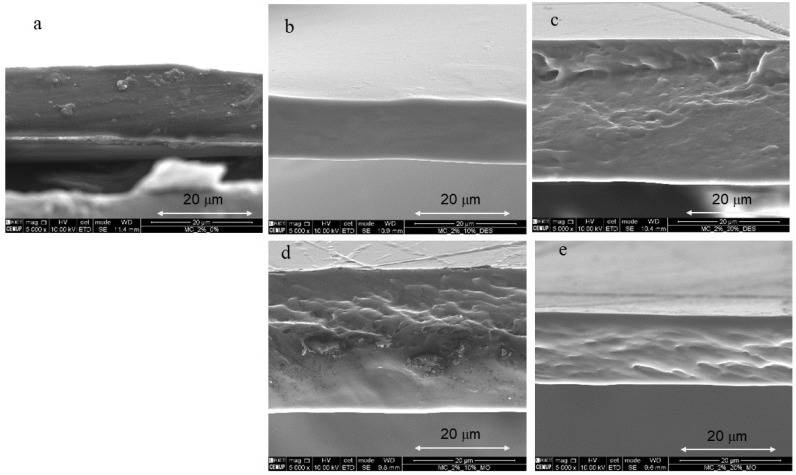
SEM images of cryo-fractured cross section (magnification 5000×, 10.00 kV). (**a**) Pure MC 2%, (**b**) MC 2%_DES 10%, (**c**) MC 2%_DES 20%, (**d**) MC 2%_MO 10%, (**e**) MC 2%_MO 20%.

**Figure 4 foods-11-02641-f004:**
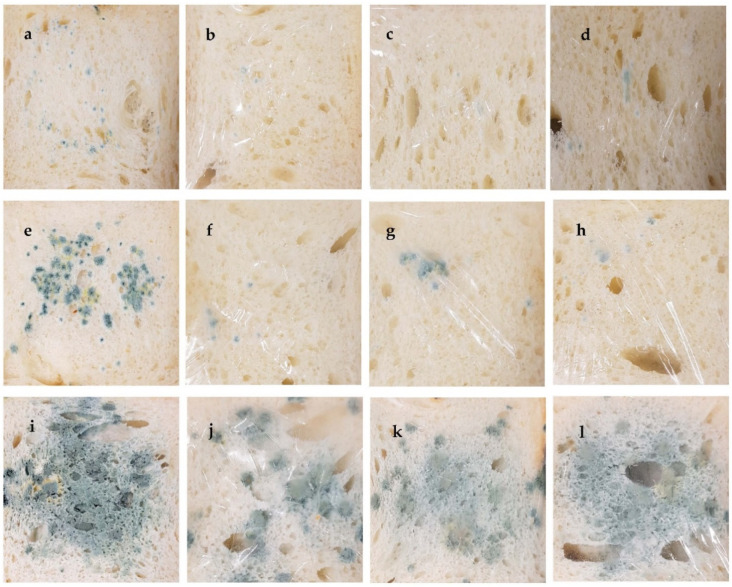
Growth of fungi naturally present in wheat bread: (**a**,**e**,**i**) control (unwrapped), (**b**,**f**,**j**) wrapped with MC 2% film, (**c**,**g**,**k**) wrapped with MC 2%_DES 10% film, (**d**,**h**,**l**) wrapped with MC 2%_MO 10% film. (**a**–**d**, after 6 days; **e**–**h**, after 7 days; **i**–**l**, after 11 days).

**Table 1 foods-11-02641-t001:** Composition of films. (wt% per dried MC amount).

Sample	Methylcellulose %	DES (ChCl:Gly, 1:2) %	MO (*Moringa oleifera* Extracted in DES) %
MC 2%	2	-	-
MC 2%-DES 10%	2	10	-
MC 2%-DES 20%	2	20	-
MC 2%-MO 10%	2	-	10
MC 2%-MO 20%	2	-	20

**Table 2 foods-11-02641-t002:** Phenolic compounds (TPC and TFC) and antioxidant capacity (DPPH, FRAP, ABTS, and ORAC).

Samples	TPC	TFC	DPPH	FRAP	ORAC	ABTS
	(mg GAE⋅g^−1^)	(mg QE⋅g^−1^)	(mmol TE⋅g^−1^)	(mmol TE⋅g^−1^)	(mmol TE⋅g^−1^)	(PI)
ChCl/LA1:2	28.314 ± 0.146 ^b^	1.595 ± 0.023 ^b^	0.176 ± 0.022 ^b^	0.469 ± 0.003 ^a^	2.107 ± 0.003 ^a^	55.529 ± 0.267 ^b^
ChCl/GLY1:2	38.409 ± 0.095 ^a^	2.259 ± 0.023 ^a^	0.294 ± 0.010 ^a^	0.361 ± 0.017 ^b^	1.200 ± 0.066 ^b^	92.614 ± 0.085 ^a^

Results are expressed as mean ± SD per gram of sample. Different superscript letters in the same column indicate significant differences at *p*
*<* 0.05.

**Table 3 foods-11-02641-t003:** Retention times, absorption maxima, and characteristic ions of tentatively identified compounds in *M. oleifera* extracts by HPLC-ESI-DAD-MS^n^.

Peak	Rt (min)	λmax (nm)	[M-H]-	MS^n^ fragments	Identification
			(*m*/*z*)	(*m*/*z*)	
1	24.6	325	353	191, 179, 135	3-Caffeoylquinic acid
2	33.1	310	337	191, 173, 163	p-Coumaroylquinicacid
3 *	40.5	325	353	191, 179, 135	5-Caffeoylquinic acid
4	52.4	271, 334	593	575, 503, 473, 383, 353	Apigenin 6,8-di-C-glucoside
5	62.5	268, 337	431	341, 311, 283	Vitexin/isovitexin
6	65.5	271, 337	431	341, 311, 283	Vitexin/isovitexin
7 *	69.1	256, 290sh, 355	609	301, 179, 151	Rutin
8	72.8	265, 290sh, 350	447	327, 285	Kaempferol-3-O-glucoside
9	73.1	253, 290sh, 355	593	285	Kaempferol-3-O-glucoside-7-rhamnoside
10	76.5	264, 289sh, 346	533	489, 447, 285	Kaempferol-3-O-malonylglucoside

* Compounds were identified by comparison with reference standards.

**Table 4 foods-11-02641-t004:** Color parameters and opacity of the different Methylcellulose films (Opacity, L*, a*, b* and ΔE*).

Material	Opacity (%)	L*	a*	b*	ΔE*
MC 2%	0.64 ± 0.07 ^a^	93.66 ± 0.20 ^a^	0.25 ± 0.04 ^a^	2.48 ± 0.18 ^a^	2.62 ± 0.09 ^a^
MC 2%_DES 10%	0.62 ± 0.06 ^b^	93.67 ± 0.06 ^a^	0.25 ± 0.03 ^a^	2.46 ± 0.18 ^a^	2.66 ± 0.04 ^a^
MC 2%_DES 20%	0.48 ± 0.03 ^c^	93.33 ± 0.14 ^b^	0.58 ± 0.09 ^b^	2.02 ± 0.06 ^b^	2.99 ± 0.14 ^b^
MC 2%_MO 10%	0.53 ± 0.02 ^d^	93.53 ± 0.14 ^a^	0.40 ± 0.02	2.14 ± 0.19 ^c^	2.78 ± 0.14 ^a^
MC 2%_MO 20%	0.97 ± 0.06 ^e^	93.25 ± 0.08 ^b^	0.53 ± 0.08 ^b^	2.06 ± 0.13 ^b,c^	3.04 ± 0.07 ^b^

Same letters in the same column indicate that values are not significantly different (*p* > 0.05).

**Table 5 foods-11-02641-t005:** Film thickness (*d*) and mechanical properties (tensile strength, TS; elongation at break, E_b_; and elastic modulus, YM) and water vapor permeability (WVP) for the methylcellulose films.

Material	d/(mm)	TS/(MPa)	Eb/(%)	YM/(MPa)	WVP∙10^−10^/ (g·m^−1^·s^−1^·Pa^−1^)
MC 2%	0.020 ± 0.001 ^a^	24.30 ± 1.7 ^a^	2.455 ± 0.31 ^a^	32.87 ± 1.0	1.40 ± 0.002
MC 2%_DES 10%	0.018 ± 0.002 ^a^	23.97 ± 1.7 ^a^	3.290 ± 0.35 ^a,b,c^	28.65 ± 0.95	1.70 ± 0.028 ^a^
MC 2%_DES 20%	0.019 ± 0.002 ^a^	20.84 ± 1.6 ^b^	4.330 ± 0.27 ^b^	22.26 ± 1.3	1.71 ± 0.074 ^a^
MC 2%_MO 10%	0.021 ± 0.002 ^a^	21.74 ± 1.3 ^c^	3.850 ± 0.79 ^c^	20.5 ± 0.47 ^a^	2.02 ± 0.039 ^b^
MC 2%_MO 20%	0.019 ± 0.001 ^a^	21.72 ± 0.71 ^b,c^	8.15 ± 0.39	18.02 ± 2.8 ^a^	2.00 ± 0.026 ^b^

Same letters in the same column indicate that values are not significantly different (*p* > 0.05).

**Table 6 foods-11-02641-t006:** Firmness of sliced bread.

Day	Bread (N·mm^−1^)			
	Control	MC	MC_DES	MC_MO
0	1.03 ± 0.05 ^a^			
1	0.97 ± 0.04 ^a^	0.96 ± 0.08	0.91 ± 0.06	1.34 ± 0.03 ^a^
4	1.59 ± 0.12	1.88 ± 0.08 ^a^	1.44 ± 0.05	1.35 ± 0.11 ^a^
7	1.94 ± 0.04 ^b^	1.94 ± 0.05 ^a^	1.58 ± 0.08	2.16 ± 0.10
11	1.99 ± 0.02 ^b^	1.95 ± 0.10 ^a^	2.14 ± 0.11	2.51 ± 0.04

Same letters in the same column indicate that values are not significantly different (*p* > 0.05).

**Table 7 foods-11-02641-t007:** Weight loss of sliced bread (%).

Sample	Day			
	1	4	7	11
Control	0.48	0.66	0.86	1.35
MC 2%	0.00	0.04	0.35	0.68
MC 2% _DES 10%	0.18	0.39	0.40	0.66
MC 2% _MO 10%	0.17	0.52	0.69	2.01

## Data Availability

Data is contained within the article or Appendix A.

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
