# Peer review of "Phenolic Extraction of Moringa oleifera Leaves in DES: Characterization of the Extracts and Their Application in Methylcellulose Films for Food Packaging"

_foods, 2022, doi:10.3390/foods11172641_

Round 1

Reviewer 1 Report

Biodegradable active packaging is important for the sustainable development. Biomass are promising materials for the design of biodegradable active packaging. The topic of this manuscript is of broad interest to the readers and the experiments are well designed. However, major revision is required.

1.     What is “NADES” in the abstract? Please give the full name.

2.     Renewable sources are promising precursors for biodegradable active packaging. Many researches have been done and more references are suggested to be cited, for example Development and Characterization of Food Packaging Bioplastic Film from Cocoa Pod Husk Cellulose Incorporated with Sugarcane Bagasse Fibre; Packaging and degradability properties of polyvinyl alcohol/gelatin nanocomposite films filled water hyacinth cellulose nanocrystals; Electrospun Functional Materials toward Food Packaging Applications: A Review.

3.     “Moringa oleifera leaf extracts, (MO)” in line 92 should be revised as “Moringa oleifera leaf extracts (MO)”.

4.     Please revise the writing of “Fe+3-TPTZ into Fe+2-TPTZ” in line 139.

5.     Please pay attention to the writing of superscripts and subscripts in line 204/207/208 and other places.

6.     Please double check the whole manuscript to solve the typos, for example “including the MCfilm.The antifungal” in line 237.

7.     Please write the units in the same way. Some are written as g·m-1·s-1·Pa-1 while some are written as TE/g.

8.     The scale bars in SEM images are hard to tell. Please replace them with much clear ones.

9.     Most of the references are too old. More references published recently are suggested to be cited.

Author Response

Reviewer #1 

Please see pdf file

Author Response

Reviewer #2

Please see pdf file

Round 2

Reviewer 1 Report

The manuscript is well revised according to the comments and could be accepted now.

Author Response

Again, we would like to thank the reviewers for the careful and thorough reading of this manuscript and the thoughtful comments and constructive suggestions, which helped improve the quality of this manuscript. 

All the best